# Concurrent Inhibition of Akt and ERK Using TIC-10 Can Overcome Venetoclax Resistance in Mantle Cell Lymphoma

**DOI:** 10.3390/cancers15020510

**Published:** 2023-01-13

**Authors:** Agnete Marie Granau, Pilar Aarøe Andersen, Theresa Jakobsen, Konstantina Taouxi, Nawar Dalila, Johanne Bay Mogensen, Lasse Sommer Kristensen, Kirsten Grønbæk, Konstantinos Dimopoulos

**Affiliations:** 1Biotech Research and Innovation Centre (BRIC), University of Copenhagen, 1165 Copenhagen, Denmark; 2Rigshospitalet, Department of Hematology, University Hospital Copenhagen, 2100 Copenhagen, Denmark; 3Department of Biomedicine, Aarhus University, 8000 Aarhus, Denmark; 4Rigshospitalet, Department of Clinical Biochemistry, University Hospital Copenhagen, 2100 Copenhagen, Denmark; 5Novo Nordisk Foundation Center for Stem Cell Biology, DanStem, Faculty of Health Sciences, University of Copenhagen, 1165 Copenhagen, Denmark

**Keywords:** mantle cell lymphoma, venetoclax, TIC-10, PI3K/Akt

## Abstract

**Simple Summary:**

Targeting BCL-2 through venetoclax is an effective therapy for a series of hematological cancers, such as mantle cell lymphoma (MCL), but resistance to venetoclax is an increasing challenge that needs to be overcome. In order to elucidate the resistance mechanisms to venetoclax in MCL at a molecular level, we used an in vitro model of acquired resistance and performed genetic, epigenetic and transcriptomic analyses. We found that venetoclax-resistant (VR) cells acquired a *TP53* mutation and consequently exhibited a reduced apoptotic response. In addition, transcriptomic analysis showed an upregulation of the PI3K/Akt pathway in the VR cells, and an extensive drug screen revealed that Akt and ERK inhibition with TIC10 could induce apoptosis in VR cells with both acquired and intrinsic venetoclax resistance. Thus, TIC-10 might be a possible treatment option for VR patients that requires further investigation.

**Abstract:**

Venetoclax, a BCL-2 inhibitor, has proven to be effective in several hematological malignancies, including mantle cell lymphoma (MCL). However, development of venetoclax resistance is inevitable and understanding its underlying molecular mechanisms can optimize treatment response. We performed a thorough genetic, epigenetic and transcriptomic analysis of venetoclax-sensitive and resistant MCL cell lines, also evaluating the role of the stromal microenvironment using human and murine co-cultures. In our model, venetoclax resistance was associated with abrogated *TP53* activity through an acquired mutation and transcriptional downregulation leading to a diminished apoptotic response. Venetoclax-resistant cells also exhibited an upregulation of the PI3K/Akt pathway, and pharmacological inhibition of Akt and ERK with TIC-10 led to cell death in all venetoclax-resistant cell lines. Overall, we highlight the importance of targeted therapies, such as TIC-10, against venetoclax resistance-related pathways, which might represent future therapeutic prospects.

## 1. Introduction

Mantle cell lymphoma (MCL) is a rare subtype of non-Hodgkin lymphoma (NHL) characterized by the t(11;14)(q13;q32) translocation that leads to constitutive overexpression of the cell-cycle-promoting protein cyclin D1 [1]. Novel treatment regimens, especially the introduction of high-dose therapy with autologous stem cell support, have significantly improved patient survival; however, MCL remains an incurable disease with late relapses, even beyond 10 years [2,3].

Venetoclax (or ABT-199) is an inhibitor of the anti-apoptotic protein BCL-2 and has recently been shown to be effective in several hematological malignancies [4]. Among NHL, venetoclax is particularly active against MCL and has been tested in several clinical trials both as a monotherapy and in combination with other drugs, such as ibrutinib [4,5,6]. However, despite the very good response rates, a major challenge is the lack of deep responses and the inevitable development of a resistant phenotype, followed by a clinical relapse.

Resistance to venetoclax has been attributed both to alterations within the malignant cells, but also their microenvironment [7,8,9]. A switch of the apoptotic dependency from BCL-2 to other anti-apoptotic proteins, such as MCL-1 or BCL-XL, or even a general reduction of apoptotic priming has been described in venetoclax resistance in several hematological malignancies [10,11,12]. Mutations in *BCL2* are uncommon in venetoclax-resistant MCL patients [13], but other genetic aberrations, such as deletion of 18q21, which includes *BCL2*, have recently been described [14]. In addition, many of the studies in MCL have investigated venetoclax and ibrutinib co-resistance [7,8,15], leaving an unmet need to determine which specific mechanisms are involved in the loss of sensitivity to venetoclax. Thus, the precise mechanisms of venetoclax resistance in MCL, including the involvement of epigenetic and/or microenvironmental factors, are yet to be deciphered.

In this study, we aimed to investigate the molecular mechanisms of acquired venetoclax resistance (VR) in MCL cell lines, as well as the effects of the microenvironment in generating a VR phenotype in MCL. In addition, we examined whether VR can be overcome by targeted therapy, based on the epigenetic and transcriptional profile of the VR cells. By analyzing the genetic, epigenetic, and transcriptomic alterations of VR MCL, we found that the *TP53* network and PI3K/Akt pathway play a central role in the development of resistance, and that pharmacological Akt/ERK inhibition with TIC-10 induced apoptosis in all VR cells.

## 2. Materials and Methods

### 2.1. Cell Cultures and Development of Venetoclax-Resistant Cell Lines

The human MCL cell lines JeKo-1, Z-138, MINO, and MAVER-1 were purchased from DSMZ (Leibniz Institute DSMZ–German Collection of Microorganisms and Cell Cultures, Braunschweig, Germany) and were grown at 37 °C with 5% CO_2_ in a humidified atmosphere and under sterile conditions using the cell culture medium recommended by DMSZ. Cell density and viability were determined using the cell counter NC-250 (Chemometec, Hillerød, Denmark). To develop venetoclax-resistant cell lines, we exposed the venetoclax-sensitive cell lines MINO and MAVER-1 to increasing doses of venetoclax (MedChemExpress, Copenhagen, Denmark) for 4–6 months, starting with their respective IC_50_ dose, as previously described [16]. The dose of venetoclax was increased gradually whenever the cells exhibited viability above 90%, reaching a final dose of 5 μm. Prior to all experiments, the venetoclax-resistant cells were left untreated for a minimum of 7 days. To control for confounding effects of prolonged cell culture, we also cultured a flask of MINO and MAVER-1 for 4–5 months, but without the addition of venetoclax.

For co-cultures of MCL cells and stromal cells, we used the human stromal cell line HS5 and the mouse stromal cell line MS5 (DMSZ, Leipzig, Germany). Briefly, when stromal cells reached a confluence of at least 80%, MCL cells were added, and after 24 h, venetoclax was added to the co-culture. After a further 48 h incubation, apoptosis was measured with flow cytometry for both the stromal cells and MCL cells.

### 2.2. Cell Proliferation and Apoptosis Assays

Cell proliferation was determined using the CellTiter 96^®^ AQueous One Solution Cell Proliferation Assay (Promega, Madison, WI, USA). Briefly, 2–4 × 10^5^ cells/mL were seeded in 96-well plates and treated with serial doses of venetoclax. After 48 h, 20 μL of MTS solution was added into each well, and following a further incubation of 3–4 h, absorbance was measured at 490 nm. Dose-response curves and calculation of IC_50_ values for venetoclax for each cell line were done using GraphPad Prism 9 software (GraphPad software, Inc. La Jolla, CA, USA).

Determination of cell apoptosis as a measure of venetoclax sensitivity of MCL cell lines in mono- and co-cultures was done using the FITC Annexin V Apoptosis Detection Kit I (BD Biosciences, San Jose, CA, USA), according to the manufacturer’s protocol. For apoptosis analysis of co-cultures, we used either CD90 (Thy1)-APC (Invitrogen, Waltham, MA, USA) for co-cultures containing HS5 or CD140a-APC (BD Biosciences, San Jose, CA, USA) for co-cultures containing MS5 to separate stromal cells from MCL cells when calculating the percentage of apoptotic cells.

Data were acquired using a FACS Calibur or FACS Fortessa (BD Biosciences, San Jose, CA, USA) and results were analyzed using the FlowJo software.

### 2.3. BH3 Profiling

For BH3 profiling, we used 5 × 10^5^ cells/mL cells in 500µL MEB2 buffer (150 mM mannitol, 10 mM HEPES-KOH, 150 mM KCl, 1 mM EGTA, 1 mM EDTA, 0.1% BSA, 5 mM succinate) with 10 µg/mL oligomycin. Cells were permeabilized in 0.002% digitonin and treated with BH3 peptides for 1 h at 25 °C in dark conditions. Loss of mitochondrial membrane potential (Ψm) was assessed using TMRE staining (54 nM), which was added at the last 30 min of incubation, as previously described [17]. The intensity of TMRE staining was measured with flow cytometry in a FACS Fortessa (BD Biosciences, San Jose, CA, USA) and results were analyzed using FlowJo software. We used the following peptides: BIM (0.5 and 1 µM), PUMA (10 µM), PUMA 2 a (10 µM), BAD (2 and 10 µM), NOXA (100 µM), w-HRK (100 µM), MS1 (20 µM) and FS1 (10 µM), which were purchased from TAG Copenhagen (Frederiksberg, Denmark), aiming for purity > 98% by HPLC/mass-spectrometry. The peptide sequences were found online in the Letai Lab recommendations for BH3 profiling (https://letailab.dana-farber.org/uploads/6/8/9/2/68921131/bh3_profiling_guide_appendix_20170814.pdf, accessed on 25 October 2021). We normalized all the acquired data to FCCP, a positive depolarization control, while DMSO was used as a negative control.

### 2.4. Nucleic Acid Extraction

Total DNA and RNA were isolated from each cell line using All Prep DNA/RNA/miRNA Universal kit and/or miRNEasy (Qiagen, Hilden, Germany) according to the manufacturer’s protocol. The quantity (260 nm) and quality (260/280 and 260/230 ratios) of the nucleic acids were evaluated with a NanoDrop-1000 (Thermo Scientific, Waltham, MA, USA).

### 2.5. Reverse Transcriptase-Quantitative PCR

First, cDNA was synthesized from 1 μg of RNA using the SuperScript III First Strand Synthesis kit (Life Technologies, Carlsbad, CA, USA) according to the manufacturer’s instructions. The qPCR was performed in duplicates in a 96-well format using 5 μL of diluted cDNA (1:10) and SYBR Green I Master Mix (Roche Diagnostics, Basel, Switzerland) in a total volume of 20 μL. Experiments were carried out with the following conditions: enzyme activation at 95 °C for 10 min, followed by 45 cycles of 95 °C for 10 s, 60 °C for 20 s and 72 °C for 30 s, and one final elongation cycle at 72 °C for 10 min, on a LightCycler^®^ 480 instrument II (Roche Diagnostics, Basel, Switzerland). Primer sequences used are shown in Appendix A. *SF3A1* and *PUM1* were used as reference genes, based on previous results [16].

### 2.6. AcceSssIble Assay

The details of this assay have previously been described [16]. Briefly, 250,000 cells per tube were centrifuged at 1500 rpm for 5 min and then resuspended in 60 μL PBS. For nuclei isolation, 1 mL of lysis buffer (10 mmol/L Tris (pH 7.4), 10 mmol/L NaCl, 3 mmol/L MgCl2, 0.1 mmol/L EDTA, 0.5% NP-40) was added, and following a 10 min incubation on ice, the lysed cells were centrifuged at 3000 rpm for 5 min at 4 °C. The nuclear pellets were resuspended in 1 mL wash buffer (10 mmol/L Tris (pH 7.4), 10 mmol/L NaCl, 3 mmol/L MgCl_2_, 0.1 mmol/L EDTA), centrifuged again at 3000 rpm for 5 min at 4 °C and then the following mix was added to each tube: 76.75 μL 1× NEB Buffer 2, 7.5 μL 10× NEB Buffer 2, 45 μL 1 mol/L sucrose, 5 μL 32 mmol/L S-adenosylmethionine (SAM), and 15 μL 4 U/mL M.SssI (or H_2_O for NoE tube). The samples were incubated at 37 °C, and after 7.5 min, the reaction was boosted by the addition of 5 μL of SAM and a further incubation of 10 min. The reaction was then stopped by the addition of 300 μL prewarmed Stop Solution (10 mmol/L Tris-HCl (pH 7.9), 600 mmol/L NaCl, 1% SDS, 0.1 mmol/L EDTA) and 3 μL Proteinase K (20 mg/mL) and a prolonged incubation at 55 °C for 16 h. DNA was then isolated by phenol/chloroform extraction and ethanol precipitation, and finally redissolved in 21 μL nuclease-free water. One μg of DNA was bisulfite-converted using the Zymo EZ DNA Methylation Kit and the efficacy of M.SssI treatment was evaluated with HRM qPCR as previously described [18], using ACTB and GRP78 as controls (Appendix A).

### 2.7. Genome-Wide Methylation and Chromatin Accessibility Analysis

We analyzed 1 µg of bisulfite-treated DNA from Acce*SssI*ble with the Infinium HumanMethylationEPIC BeadChip array (Illumina, Inc., San Diego, USA). Data import of idat files, normalization, background and dye bias correction, and calculation of β values were performed using the R package *minfi* [19]. Probes containing a SNP (N = 340,327), cross-reactive probes (N = 42,558) (both based on [20]), probes located on the sex chromosomes (N = 19,681), and probes with a high detection *p*-value (above 0.01) were removed before further analyses.

Accessibility (Acc) was calculated as the difference between β values of the M.SssI-treated sample and the NoE sample. We then calculated the difference in accessibility (ΔAcc) and methylation (ΔMeth) between resistant and sensitive cell lines by subtracting the Acc and NoE values of the venetoclax-resistant cell lines from the sensitive ones. ΔMeth and ΔAcc values of ±0.15 for each analyzed CpG site were considered as significant.

### 2.8. Magnetic Cell Sorting of MCL Cells from Co-Cultures for RNA Extraction

Co-cultured MCL cells were first separated into layer- and supernatant-located cells and incubated for 15–30 min with either CD90-APC (HS5 co-cultures) or CD140a-APC (MS5 co-cultures). After washing, 50µL APC Magnetic Particles (BD Biosciences, San Jose, CA, USA, catalog no. 557932) was added to each sample and followed by another incubation at 4 °C for 30 min. Then, 1× BD iMAG buffer (BD Bioscienses) was added to a final maximum volume of 1 mL and the tubes were placed on a magnetic separation column for 8 min. Purity was measured on the BD FACS Calibur. In samples with purity below 90%, sorting was repeated with an additional incubation with the magnetic particles, until purity was above 90%.

### 2.9. Whole Exome Sequencing

Whole Exome Sequencing (WES) was performed by BGI using the DNBSEQ platform and with 1µg of genomic DNA for each sample. Fastq files were analyzed according to the Broad Institute overview and GATK (Genome Analysis Tool Kit) Best Practice documentation. The workflow was written in Workflow Description Language (WDL) format v1.0 and run using Cromwell v72 as the workflow management system. Reads were aligned to the hg38 human genome and somatic variants were called for each sample using GATK4 Mutect2, comparing its genome with a pool of healthy, non-tumor genomes (https://gatk.broadinstitute.org/hc/en-us/articles/360035531132, accessed on 14 March 2022).

The generated VCF files were uploaded onto BCFtools (v1.15) (http://github.com/samtools/bcftools, accessed on 14 March 2022) that was used to further filter out variants present in the parental and long-term culture without venetoclax addition (background variants). The remaining variants were then deemed relevant if they had a GnomAD score below 0.001 and an estimated HIGH Impact based on the Ensembl Variation–Calculated variant consequences tool. Finally, we used the Ensembl VEP tool (https://www.ensembl.org/Tools/VEP, accessed on 14 March 2022) to annotate the detected variants.

### 2.10. Transcriptome Analysis

A total of 1 μg of RNA was used for RNA-seq analysis. The sample quality control, library preparation and quality control, sequencing and data quality control were performed by BGI. Library preparation was done using ribosomal depletion and RNA was then fragmented and reverse-transcribed to double-stranded cDNA (dscDNA) using random hexamer primers. Each library was then sequenced at a depth of 50 m clean reads using the DNBSEQ platform.

Raw reads were adapter-trimmed using trim_galore v0.6.6. Reads from potential ribosomal RNA left over from the ribodepletion step were first removed using bbsplit (bbtools v37.62). Then, the remaining reads from the bbsplit output were mapped to the human genome (hg19) using STAR v2.7.7a (default settings). Mapped reads were quantified using featureCounts v2.0.1 (default settings), and differential gene expression was determined using the R package *DESeq2* [21]. Transcripts were considered significant based on both their fold-change (absolute value of log2-fold change above 1) and their adjusted *p*-value using the Benjamini–Hochberg method (*p* < 0.05). Heatmaps and PCA plots were generated using R (v3.6.1).

### 2.11. Gene Set Enrichment Analysis

Gene Set Enrichment Analysis (GSEA) was done using the R package *gprofiler2*. Gene sets used for the analysis of both Acce*SssI*ble and transcriptomic data were the KEGG subset, REACTOME subset, GO: Gene Ontology and WikiPathways gene sets. Selected pathways were chosen based on their adjusted *p*-values (*p* < 0.01) and biological relevance. The query-gene lists used up- and downregulated genes based on the Acce*SssI*ble analysis; differentially expressed genes between venetoclax-sensitive and resistant cell lines and up- and downregulated genes from the two co-cultures.

### 2.12. Statistical Analysis

All the statistical analyses for Acce*SssI*ble, WES, and RNA-seq were performed in the statistical software R (v3.6.1). For the comparison of mean values of different groups, we used ANOVA (assuming that data were normally distributed for each group) in GraphPad Prism 9 software. When multiple comparisons were performed, the *p*-values were corrected with the Bonferroni method. The level for significance was set at *p* < 0.05 with the following annotation: * *p* < 0.05, ** *p* < 0.01, *** *p* < 0.001 and **** *p* < 0.0001.

## 3. Results

### 3.1. Generation of In Vitro Models of Acquired and Microenvironmentally Induced Venetoclax Resistance

Initially, we established a model of acquired venetoclax resistance, using the two venetoclax-sensitive MCL cell lines MINO and MAVER-1. After 4–6 months of continuous exposure to gradually increasing doses of venetoclax up to 5 µM, these cell lines became venetoclax-resistant (VR) (denoted MINO-VR and MAVER1-VR). To control for the effects of long-term (LT) culture, we cultured MINO and MAVER-1 cells for the same amount of time (denoted MINO-LT and MAVER1-LT). Prolonged culture did not affect the venetoclax sensitivity of MINO cells; however, MAVER1-LT responded to a lesser extent to venetoclax, albeit not exhibiting a fully resistant phenotype (Figure 1a). The IC50 of venetoclax for all the cell lines can be found in Table 1.

To evaluate the role of microenvironment-induced resistance against venetoclax, we co-cultured MINO and MAVER-1 with both human stromal cells (HS5) and murine stromal cells (MS5). However, while MCL cells co-cultured with MS5 exhibited a venetoclax-resistant phenotype, co-culture with HS5 failed to abrogate venetoclax toxicity (Figure 1b). Notably, we did not see the same difference between HS5 and MS5 co-cultures when testing other chemotherapeutic agents, such as doxorubicin and vincristine (Appendix A). To further examine the protective effect of MS5 during exposure to venetoclax, we analyzed both the adherent fraction of MCL cells in direct contact with the stromal cells and MCL cells from the suspension in the co-culture separately. We found that the stromal-mediated venetoclax resistance in MS5 co-cultures was mainly observed in the MCL cells in suspension and not the adherent cells (Figure 1c).

To determine whether the MS5-induced resistance was mediated through a soluble factor, we cultured MCL cells in MS5-conditioned medium prior to treatment with venetoclax. Interestingly, culturing the cells in conditioned medium from MS5 also resulted in resistance to venetoclax, albeit to a lesser extent (Appendix A). Additionally, we found that removing the MCL cells from the MS5 co-culture and resuspending them in fresh medium restored their sensitivity to venetoclax (Appendix A), suggesting that microenvironment-induced venetoclax resistance is only transient.

### 3.2. Venetoclax Resistance Is Associated with Changes in BH3-Mediated Apoptotic Priming

We first determined whether the resistant phenotype was confined to venetoclax or whether it also affected other anti-apoptotic BH3 family members, such as MCL-1 or BCL-X_L_. The MINO-VR cells also showed decreased sensitivity to the MCL-1 inhibitor S63845 (while MAVER-VR were slightly more sensitive to S63845), as well as the BCL-X_L_ inhibitor WEHI-539, although resistance to BCL-X_L_ inhibition was evident already in the parental MAVER-1 (Figure 2A and Appendix A). This combined BH3 family resistance was also observed in the co-culture-mediated venetoclax resistance. Treatment with 1 µM S63845 and 1 µM WEHI-539 led to a significantly lower apoptotic response in MINO and MAVER co-cultured with MS5 (Figure 2B). Thus, both acquired and induced venetoclax resistance were not specific for venetoclax, but potentially associated with reduced apoptotic priming to all three antiapoptotic BH3 proteins.

To further evaluate these findings and determine the overall apoptotic priming of sensitive and resistant cells, we performed BH3 profiling. This method reveals the degree of apoptotic deficiency of the cells to non-specific apoptotic activators (such as BIM, PUMA) or BH3 protein-specific sensitizers (for example BAD for BCL-2, NOXA for MCL-1, and HRK for BCL-X_L_). Consistent with their venetoclax resistance, both VR cell lines exhibited a significantly reduced apoptotic response to BAD, a sensitizer that primarily binds to BCL-2 (Figure 2C). In addition, we observed a significant decrease in the overall apoptotic sensitivity of VR cells, particularly to the pan-activator PUMA, while MINO-VR also showed a partially abrogated response to BIM (Figure 2C). Notably, MINO showed a significant response to HRK, suggesting a high apoptotic dependency on BCL-XL, which was not observed in MINO-VR, a finding that confirms our previous data showing loss of sensitivity to the BCL-XL inhibitor WEHI-539 in MINO-VR. Thus, BH3 profiling showed that acquired venetoclax resistance was associated with decreased dependency to BCL-2; however, this occurred without a consequent increased dependency to either MCL-1 or BCL-X_L_, but rather through an overall reduced apoptotic response.

### 3.3. Acquired Venetoclax Resistance Is Associated with Very Few Genetic Events but an Extensive Epigenetic Reprogramming

To explore whether acquired venetoclax resistance might be genetically driven, we performed whole-exome sequencing and compared the genome of the VR cells to the sensitive cells for any novel genetic alterations. Our analysis revealed a total of 21 overlapping mutations between the two VR cell lines (Appendix A). Notably, we found no genetic abnormalities affecting *BCL2* or other genes of the BH3 family. However, we found that both VR cells had acquired a frameshift *TP53* mutation (Appendix A), which is an additional “hit” since both parental MINO and MAVER-1 are already known to carry a *TP53* mutation.

We then evaluated the role of epigenetic changes during venetoclax resistance by simultaneously measuring DNA methylation and chromatin accessibility across the genome using Acce*SssI*ble, an assay based on the Illumina 850 K EPIC array. We discovered numerous probes with changes in both DNA methylation and chromatin accessibility in VR cells, while long-term cultured cells without venetoclax had an epigenetic profile similar to the respective parental cells (Figure 3a,b). Interestingly, none of the promoters of *BCL2*, *MCL-1* or *BCL2L1* (coding for BCL-X_L_) contained altered patterns of DNA methylation or chromatin accessibility (Appendix A).

To confirm the importance of these findings, we treated the VR cell lines with the DNA methyltransferase inhibitor 5-azacytidine (a hypomethylating agent) and the EZH2 inhibitor EPZ- 6438. Based on previous data [16], we used 5-azacytidine and EPZ-6438 for 48 h, both alone and in combination, before exposing the cells to venetoclax. However, none of the epigenetic drugs were effective in reversing venetoclax resistance (Appendix A). Similarly, the addition of 0.1 µM of 5-azacytidine in the MS5 co-cultures 24 h prior to venetoclax treatment did not alter the MS5-induced venetoclax resistance (Appendix A). Overall, our data suggest that epigenetic modifications might be secondary and not the main driver of venetoclax resistance.

### 3.4. Epigenetic and Transcriptomic Changes Highlight the Role of the PI3K/Akt Pathway in Venetoclax Resistance

To further evaluate the impact of the epigenetic changes in VR cells, we measured the transcriptomic changes of VR cells with RNA sequencing (RNA-seq). Principal component analysis (PCA) showed that the transcriptome of VR cells was extremely convergent, in strong contrast with the parental cell lines, whose transcriptome was very distinct from each other and closer to the cells cultured long-term without venetoclax (Figure 3c,d). Differential gene expression analysis showed a total of 1913 differentially expressed genes between the VR and parental cell lines; 923 upregulated and 990 downregulated (Figure 3e). We next examined whether there was any overlap between the observed epigenetic changes and transcriptomic changes. After locating promoter-related probes, we defined them as “upregulated” if they exhibited significantly increased chromatin accessibility and/or decreased DNA methylation, and “downregulated” when vice versa. We then evaluated the overlap between these epigenetically changed promoters with the differentially expressed genes from our RNA-seq analysis. Interestingly, approximately 39% of the upregulated genes and 46% of the downregulated genes were found to have epigenetic changes correlating with active and reduced transcriptional activity, respectively, in their promoters (Figure 3f).

We then analyzed the upregulated and downregulated genes using gene set enrichment analysis (GSEA) to discover pathways implicated in venetoclax resistance. Notably, for the upregulated genes, we observed a significant enrichment of genes involved in the PI3K/Akt pathway, signaling by G protein-coupled receptors (GPCR) and tyrosine kinase signaling, all of which are potentially druggable (Figure 3g). On the other hand, the downregulated genes showed significant enrichment of apoptotic pathways, the JAK/STAT signaling pathway, and regulatory networks of *TP53* (Figure 3h). We also performed GSEA for the gene promoters with significant changes in DNA methylation and/or chromatin accessibility and found three overlapping pathways among the epigenetically upregulated promoters and upregulated transcripts from RNA-seq (including the PI3K/Akt signaling pathway, Figure 3f); however, there were no overlapping pathways among the downregulated genes and the epigenetically silenced promoters.

Finally, we also performed RNA-seq in MCL cells from co-cultures after separating the MCL cells into two fractions: cells adhered to the stromal cells and cells in suspension. Both fractions were sorted from the stromal cells using magnetic beads. However, since venetoclax resistance was mainly observed in the suspension MCL cells, we only included those in the subsequent analyses. In strong contrast to the VR cells, RNA-seq from co-cultures showed only a total of 107 differentially expressed genes; 105 were downregulated and only two upregulated (Figure 3i). Thus, GSEA for the upregulated genes from co-culture experiments was not feasible. However, GSEA of the downregulated genes revealed a degree of overlap with the VR cells, involving pathways such as the JAK/STAT and TNF signaling pathways (Figure 3g).

### 3.5. Pharmacological Inhibition of Akt and ERK Is Toxic to Venetoclax-Resistant Cells

We then aimed to identify compounds that could potentially restore venetoclax sensitivity. For that, we exposed both the paternal and VR cell lines for 48 h to a total of 13 different compounds, especially focusing on the PI3K/Akt pathway (Appendix A), followed by either no further treatment or venetoclax for an additional 48 h. Interestingly, none of the direct PI3K or Akt inhibitors exhibited a resensitizing effect; however, we found that TIC-10, a combined Akt and ERK inhibitor and activator of the endogenous tumor-suppressor gene tumor necrosis factor–related apoptosis-inducing ligand (*TRAIL*), was particularly toxic for both VR cell lines, even without the presence of venetoclax (Figure 4a). Furthermore, we found that in addition to the VR cell lines, two cell lines with intrinsic venetoclax resistance, JeKo-1 and Z-138, were more sensitive to TIC-10 than the venetoclax-sensitive cell lines (Figure 4b).

Finally, we measured the expression changes of the BH3 family members using qPCR before and after treatment with TIC-10. Interestingly, there was no upregulation of the pro-apoptotic family members BIM, PUMA or NOXA during TIC-10 treatment, apart from MINO-VR, where all three transcripts were upregulated following 72 h of treatment with TIC-10 (Figure 4c). This finding suggests that the apoptotic response of VR cells to TIC-10 might not be entirely BH3-dependent and that treatment with TIC-10 might be able to overcome the reduced apoptotic status of MINO-VR and MAVER-VR.

## 4. Discussion

The development of chemotherapy resistance remains a problem confronting clinicians, especially in an era of a constantly changing therapeutic landscape. Accordingly, venetoclax resistance has been reported in both myeloid and lymphoid malignancies [7,12,14,22,23,24,25,26], and understanding its underlying mechanisms may allow for the development of strategies to overcome it. In this study, we explored the molecular mechanisms of venetoclax resistance in MCL using two separate models: acquired resistance, using continuous exposure to venetoclax, and microenvironmentally induced resistance, using co-cultures with human and murine stromal cells.

Surprisingly, we found that VR cells did not exhibit a “switch” of apoptotic dependency from BCL-2 to either MCL-1 or BCL-X_L_. In fact, VR cells were less dependent on all the antiapoptotic BH3 proteins and, in addition, they exhibited an overall decrease in their apoptotic response. This finding strongly contrasts with previous studies that have shown that increased levels of MCL-1, as well as an increased apoptotic dependency to MCL-1, have been described as the main driver mechanisms of venetoclax resistance in CLL [25,27], diffuse large B-cell lymphoma [24,28], multiple myeloma [12,29], and acute myeloid leukemia [11,26]. Our data implies that patients with venetoclax resistance in MCL might not benefit from additional inhibition of MCL-1 or BCL-X_L_.

Mutations and deletions of *BCL2* or other members of the BH3 family, such as *BAX*, have previously been described in VR CLL clones [10,14,23], but were not detected in our VR cells. However, we found a newly acquired frameshift mutation in *TP53* that was present in both VR cell lines, and which further resulted in a drastic fall of *TP53* transcript levels. Previous studies have also revealed *TP53* mutations in venetoclax-resistant CLL and other NHLs [24,30], but its precise role in the resistant phenotype has yet to be deciphered. Since PUMA-mediated apoptosis requires the nuclear presence of p53 [31], we hypothesize that the abrogated function of p53 might be essential for the reduced apoptotic priming of venetoclax-resistant cells. Indeed, one of the most striking findings of BH3 profiling in our VR cells was their reduced apoptotic response to PUMA, which further supports this hypothesis.

In our data, the transcriptional changes of the VR cells were extremely convergent, and gene ontology further emphasized the potential involvement of the PI3K/Akt pathway in venetoclax resistance. This finding is in line with a recent study which also showed dysregulation of the PI3K/Akt pathway in venetoclax resistance in MCL [32]. A metabolic reprogramming with involvement of Akt has previously also been reported in VR cells in other hematological cancers, and pharmacological inhibition of Akt has been shown to restore venetoclax sensitivity [24,25,26,28]. On the other hand, RNA-seq analysis of the MS-5 co-cultured cells only revealed two upregulated genes: *WNT10A* and *IGFBP2,* thus not allowing a GSEA analysis. Interestingly, it has previously been shown that IGFBP-2 is regulated through PI3K/Akt in both malignant and non-malignant cells [33,34], while increased levels of IGFBP-2 have also been associated with a chemotherapy-resistant phenotype in AML [35].

When we treated our VR cells with the Akt/ERK inhibitor TIC-10, we did not find synergy or resensitization to venetoclax, but, rather unexpectedly, a direct cytotoxic effect, which we also observed in the two intrinsically VR cell lines (Jeko-1 and Z-138). In particular, the role of Akt has previously been highlighted in VR lymphomas, where both knock-down and pharmacological inhibition of Akt resensitized cells to venetoclax [22]. However, in our drug screen, the Akt-inhibitor Capivasertib did not induce apoptosis as effectively as TIC-10, potentially highlighting the importance of both Akt and ERK in venetoclax resistance in MCL. Surprisingly, isolated PI3K inhibition or even isolated Akt inhibition were not sufficient to induce apoptosis in the VR cells. We speculate that there must be some form of crosstalk and redundancy between the PI3K/Akt/mTOR and MAPK/ERK pathways. Thus, it might be possible that inhibition of a single component of the PI3K/Akt/mTOR pathway fails to kill VR cells, because the MAPK/ERK pathway can partially maintain some of the PI3K/Akt/mTOR pathway transcriptional network. In addition, TIC-10 has been shown to upregulate the expression of the endogenous tumor-suppressor gene *TRAIL*, which is a potent apoptosis inducer, in a p53-independent manner [36]. Deregulation of TRAIL has recently been reported in venetoclax resistance in CLL [24] and AML [26], and our data support that it might also be associated with VR in MCL.

Even though we extensively examined both acquired and microenvironmentally induced venetoclax resistance in MCL, we only used in vitro models, based on two venetoclax-sensitive cell lines. Despite that, we found that these cell lines shared genetic and transcriptional landscapes upon developing venetoclax resistance. Further studies, especially including primary patient samples, are needed to confirm the translational value of our findings. In addition, even though we do not provide a precise mechanistic model, we find that the combined inhibition of Akt and ERK, using TIC-10, is especially toxic for VR cells, potentially due to abrogation of the PI3K/Akt pathway that is responsible for the resistance to venetoclax.

## 5. Conclusions

In conclusion, our study identified complex mechanisms underlying acquired resistance to venetoclax in MCL. Our data suggest a general apoptotic reluctancy, possibly due to *TP53* mutations, in VR cells, as well as a crucial role of PI3K/Akt in the VR phenotype. Inhibition of Akt and ERK was particularly effective in both intrinsic and acquired venetoclax resistance, and therapies specifically targeting the PI3K/Akt pathway, such as TIC-10, may represent future directions for VR MCL.

## Figures and Tables

**Figure 1 cancers-15-00510-f001:**
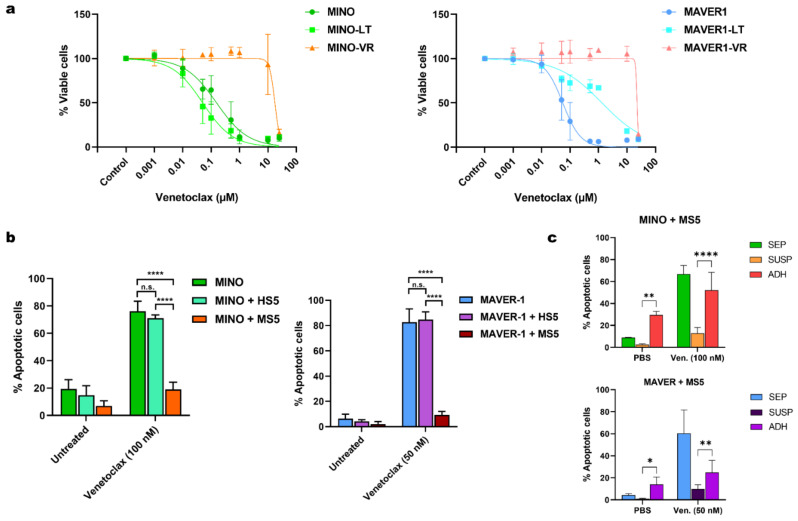
(**a**) Dose-response curves for venetoclax and MINO, (left), MAVER-1 (right). (**b**) Bar charts showing the apoptotic response to venetoclax in co-cultures of MINO (left) and MAVER-1 (right) with HS-5 and MS-5. Data are presented as mean + SD (*n* = 3). (**c**) Bar charts with the mean apoptotic response to venetoclax in MCL cells without co-culture (SEP), the co-cultured MCL cells in suspension (SUSP) and the adherent MCL cells in co-culture ADH), shown for MINO (upper) and MAVER-1 (lower) in MS-5 co-cultures. Data are presented as mean + SD (*n* = 3). Abbreviations: MINO-LT: long-term cultured MINO; MINO-VR: venetoclax-resistant MINO; MAVER1-LT: long-term cultured MAVER-1; MAVER1-VR: venetoclax-resistant MAVER-1; SD: standard deviation. * *p* < 0.05, ** *p* < 0.01, and **** *p* < 0.0001.

**Figure 2 cancers-15-00510-f002:**
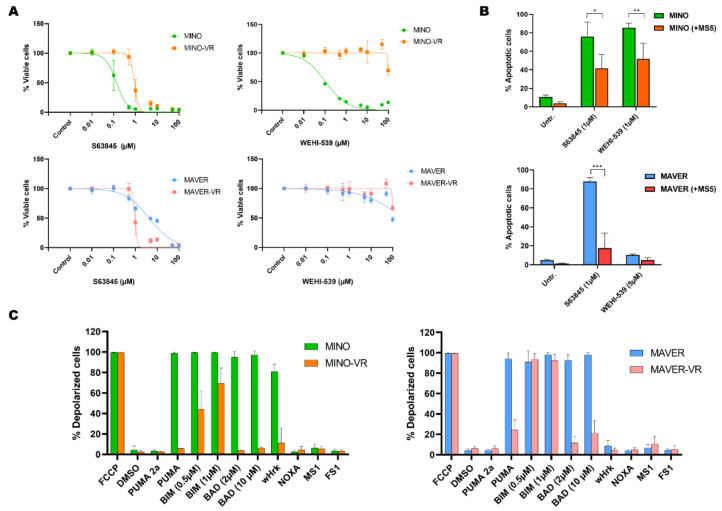
(**A**) Dose-response curves for the MCL-1 inhibitor S63845 and the BCL-XL inhibitor WEHI-539 in MINO/MINO-VR (upper panels), and MAVER/MAVER-VR (lower panels). (**B**) Bar charts of the apoptotic response to S63845 in co-cultures of MINO (upper) and MAVER-1 (lower) with MS-5. Data are presented as mean + SD (*n* = 3). (**C**) Bar charts with a summary of BH3 profiling data for MINO/MINO-VR (left) and MAVER/MAVER-VR (right) in MS-5 co-cultures. Data are presented as mean + SD (*n* = 3). Abbreviations: MINO-VR: venetoclax-resistant MINO; MAVER-VR: venetoclax-resistant MAVER-1. * *p* < 0.05, ** *p* < 0.01 and *** *p* < 0.001.

**Figure 3 cancers-15-00510-f003:**
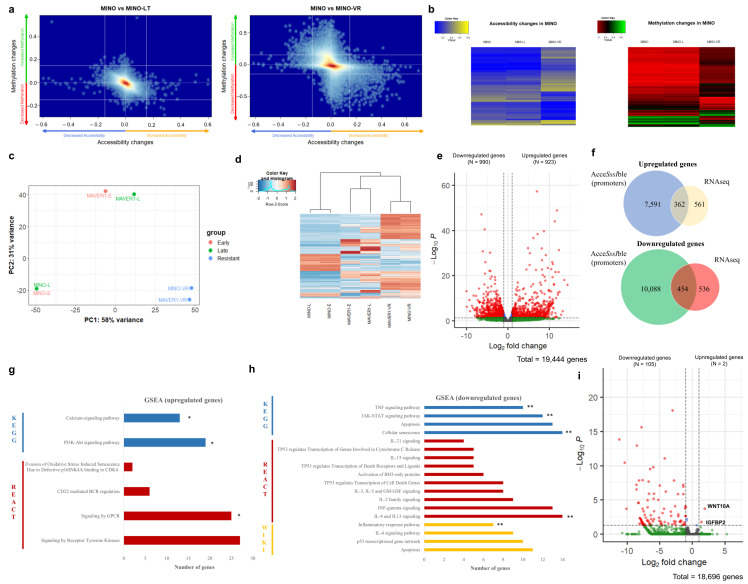
(**a**) Kernel density plot of the combined DNA methylation changes and chromatin accessibility changes observed in MINO-LT (left) and MINO-VR (right) cell lines in comparison to the paternal MINO cell line. Each dot represents an Illumina probe, delta-accessibility and delta-methylation of the corresponding comparison, with a significant difference threshold of ±15%. (**b**) Heatmaps of raw accessibility (left) and DNA methylation (right) values for MINO, MINO-LT and MINO-VR for all Illumina probes with an accessibility and/or methylation difference of ±15% compared to the paternal MINO. (**c**) Principal component analysis (PCA) plot from RNA-seq data for all cell lines, depicting their relative transcriptome proximity. (**d**) Heatmap with unsupervised clustering including the 500 most variable genes among all cell lines. (**e**) Volcano plot with the differentially expressed genes between the VR cell lines and paternal cell lines (dark grey: no significant change, green: absolute log2-fold change > 2, but adjusted *p*-value > 0.05, red: absolute log2-fold change > 2, and adjusted *p*-value ≤ 0.05). (**f**) Venn diagrams showing the overlap between upregulated Acce*SssI*ble promoters and upregulated genes by RNA-seq (upper) and between downregulated Acce*SssI*ble promoters and downregulated genes by RNA-seq (lower). (**g**) Gene set enrichment analysis (GSEA) for the upregulated genes from RNA-seq analysis. The asterisks show the overlapping pathways from GSEA using the upregulated Acce*SssI*ble promoters. (**h**) Gene set enrichment analysis (GSEA) for the downregulated genes from RNA-seq analysis. The double asterisks demonstrate the overlapping pathways from GSEA using the downregulated genes from RNA-seq analysis of MS-5 co-cultures (microenvironment-induced resistance). (**i**) Volcano plot with the differentially expressed genes between the MS5 co-cultured cell lines and HS-5 co-cultured cell lines (dark grey: no significant change, green: absolute log2-fold change > 2, but adjusted *p*-value > 0.05, red: absolute log2-fold change > 2, and adjusted *p*-value ≤ 0.05). * *p* < 0.05 and ** *p* < 0.01.

**Figure 4 cancers-15-00510-f004:**
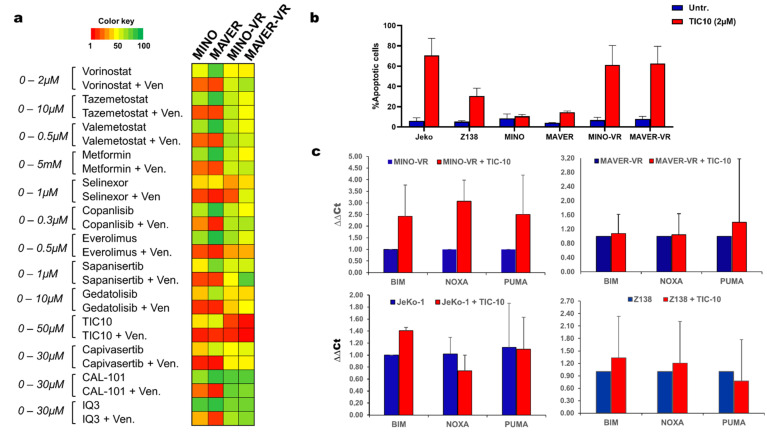
(**a**) Heatmap with the area under the curve (AUC) values for the drug-screen results using 13 different compounds with and without venetoclax for all cell lines. (**b**) Bar chart depicting the apoptotic response following treatment with 2 µM TIC-10 for 72 h. Data are shown as mean + SD (*n* = 3). (**c**) Bar charts with the relative expression of the pro-apoptotic transcripts BIM, PUMA and NOXA, before and after treatment with 2 µM TIC-10 for 72 h, measured with qPCR and analyzed with the ∆∆Ct method. Data are shown as mean + SD (*n* = 3).

**Table 1 cancers-15-00510-t001:** IC_50_ values for venetoclax across all cell lines, as calculated by dose-response curves.

Cell Line	Venetoclax IC50 (µM)
*Parental cell lines*	
JeKo-1	5.003
Z-138	6.071
MINO	0.154
MAVER-1	0.055
*Long-term (LT) cultured cell lines*	
MINO-LT	0.050
MAVER1-LT	1.320
*Venetoclax-resistant (VR) cell lines*	
MINO-VR	17.55
MAVER1-VR	22.63

## Data Availability

The authors declare that all data supporting the findings of this study are available within the paper and raw data can be provided freely upon request to the corresponding author(s).

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
