# Peer review of "Concurrent Inhibition of Akt and ERK Using TIC-10 Can Overcome Venetoclax Resistance in Mantle Cell Lymphoma"

_cancers, 2023, doi:10.3390/cancers15020510_

Round 1
Reviewer 1 Report
In this manuscript, authors explored the molecular mechanisms of venetoclax resistance in MCL, including acquired resistance and the microenvironmentally induced resistance. Genetic, epigenetic and transcriptomic analysis revealed the acquired TP53 mutation and upregulation of the PI3K/Akt signaling of these resistant models. Finally, in vitro screen demonstrated that the Akt/MEK dual inhibitor TIC-10 induced dramatic cell death in these venetoclax resistant cells.
The comments are as following:
· Resistance mechanisms Interrogation only from the 2 in vitro cell lines is weak
· PI3K/Akt signaling has already been reported to be associated to the venetoclax resistance in MCL, which was published in Am J Cancer Res. 2022;12(3):1102-1115.
· Authors did not clarify why the dual Akt/MEK inhibitors is better than the PI3K inhibitors in these resistant cells.
· In figure 2A, authors described that VR cells showed decreased sensitivity to both S63845 and WEHI-539, while this is not true for S63845 in Maver-VR cells.
· In figure 4c, authors stated that there was no upregulation of BIM, NOXA and PUMA, while all three pro-apoptotic members were clearly shown upregulated in Mino-VR cells.
· The authors should carefully check the concise scientific terms and avoid the typo errors.
e.g. JeKo-1 (Jeko1 in table 1); duplicate figure 3G legends; MCL-1 (MCL1)
· The figure quality needs to be improved
Reviewer 2 Report
The aim of this paper was to investigate the underlying molecular mechanisms of venetoclax resistance (VR) in 2 MCL cell lines gained as a result of either continuous exposure to venetoclax or to stromal microenvironment. Genetic, transcriptomic and epigenetic analysis was undertaken to understand the molecular mechanisms of venetoclax resistance. Genetic analysis revealed a newly acquired frameshift mutation in TP53 and subsequent reduction in TP53 transcript level. The major mechanism claimed for the decreased apoptotic response in VR is the frameshift mutation in TP53 and its transcriptional downregulation. Increased activity PI3K-AKT pathway was observed in the VR cells and challenged by a dual inhibitor of Akt and MEK (TIC-10) leading to induction of apoptosis in VR cells.
The MS is well written, and results are nicely presented.
I have following major and minor comments and questions:
1. The MS needs a clear explanation of the significance of the additional TP53 mutation in both VR cells which already possess missense/loss of function mutation in TP53 (MINO: V147G and MAVER-1 D281E). How was the transcript level of p53 measured? Was it confirmed by RT-PCR?
2. In the methods section, the following details are required: What is the source of the peptides? It is also not specified whether flow cytometry or a fluorimeter was used for BH3 profiling.
3. Figure 2 A: Looking at the graph for MAVER1 treated with MCL-1 inhibitor, it appears that the MAVER-VR cells are more sensitive to S63845. Could the authors upload a table of the IC-50 values in the supplementary for figure 2A as well.
4. Please check the numbering for supplementary tables. They are wrongly labeled in the supplementary file.
5. Venetoclax is misspelled in results section 3.5 line number 7.
